# The assessment of acute chorioretinal changes due to intensive physical exercise in young adults

**Irén Szalai**[1], **Anita Csorba**[1], **Fanni Pálya**[1], **Tian Jing**[2], **Endre Horváth**[3], **Edit Bosnyák**[4], **István Györe**[4], **Zoltán Zsolt Nagy**[1], **Delia Cabrera DeBuc**[2], **Miklós Tóth**[4,5], **Gábor Márk Somfai**[1,6,7] *

1 Department of Ophthalmology, Semmelweis University, Budapest, Hungary, 2 Miller School of Medicine, Bascom Palmer Eye Institute, University of Miami, Miami, FL, United States of America, 3 Independent statistician, Budapest, Hungary, 4 Department of Health Sciences and Sport Medicine, University of Physical Education, Budapest, Hungary, 5 Department of Laboratory Medicine, Semmelweis University, Budapest, Hungary, 6 Department of Ophthalmology, Stadtspital, Zürich, Switzerland, 7 Spross Research Institute, Zürich, Switzerland

* somfaigm@yahoo.com

**Data Availability Statement:** All relevant data are within the paper and its Supporting Information files.

## Abstract

### Purpose

There is abundant evidence on the benefits of physical activity on cardiovascular health. However, there are only few data on the acute effects of physical exercise on the retina and choroid. Our aim was the in vivo examination of chorioretinal alterations following short intense physical activity by spectral domain optical coherence tomography (SD-OCT).

### Methods

Twenty-one eyes of 21 healthy, young subjects (mean age 22.5 ± 4.1 years, 15 males and 6 females) were recruited. Macular scanning with a SD-OCT was performed before and following a vita maxima-type physical strain exercise on a rowing ergometer until complete fatigue. Follow-up OCT scans were performed 1, 5, 15, 30 and 60 minutes following the exercise. The OCT images were exported and analyzed using our custom-built OCTRIMA 3D software and the thickness of 7 retinal layers was calculated, along with semi-automated measurement of the choroidal thickness. One-way ANOVA analysis was performed followed by Dunnett post hoc test for the thickness change compared to baseline and the correlation between performance and thickness change has also been calculated. The level of significance was set at 0.001.

### Results

We observed a significant thinning of the total retina 1 minute post-exercise (-7.3 ± 0.6 μm, $p < 0.001$) which was followed by a significant thickening by 5 and 15 minutes (+3.6 ± 0.6 μm and +4.0 ± 0.6 μm, respectively, both $p < 0.001$). Post-exercise retinal thickness returned to baseline by 30 minutes. This trend was present throughout the most layers of the retina, with significant changes in the ganglion cell–inner plexiform layer complex, (-1.3 ± 0.1 μm, +0.6 ± 0.1 μm and +0.7 ± 0.1 μm, respectively, $p < 0.001$ for all), in the inner

**Funding:** Funding: This study was supported in part by grants from the National Research, Development and Innovation Fund of Hungary (2020-4.1.1-TKP2020 and TKP2020-NKA-17; https://nkfih.gov.hu/for-the-applicants) to M.T, the National Institute of Health (No. P30-EY014801, https://www.nih.gov/grants-funding) and the Research to Prevent Blindness, Inc. (https://www.rpbusa.org/rpb/grants/) both to University of Miami (D.C.D.). The funders had no role in study design, data collection and analysis, decision to publish, or preparation of the manuscript.

**Competing interests:** The authors have declared that no competing interests exist.

nuclear layer at 1 and 5 minutes (-0.8 ± 0.1 μm and +0.8 ± 0.1 μm, respectively, $p$ <0.001 for both), in the outer nuclear layer–photoreceptor inner segment complex at 5 minute (+2.3 ± 0.4 μm, $p$ <0.001 for all) and in the interdigitation zone–retinal pigment epithelium complex at 1 and 15 minutes (-3.3 ± 0.4 μm and +1.8 ± 0.4 μm, respectively, $p$ <0.001 for both). There was no significant change in choroidal thickness; however, we could detect a tendency towards thinning at 1, 15, and 30 minutes following exercise. The observed changes in thickness change did not correlate with performance. Similar trends were observed in both professional and amateur sportsmen (n = 15 and n = 6, respectively). The absolute changes in choroidal thickness did not show any correlation with the thickness changes of the intraretinal layers.

## Conclusions

Our study implies that in young adults, intense physical exercise has an acute effect on the granular layers of the retina, resulting in thinning followed by rebound thickening before normalization. We could not identify any clear correlation with either choroidal changes or performance that might explain our observations, and hence the exact mechanism warrants further clarification. We believe that a combination of vascular and mechanic changes is behind the observed trends.

## Introduction

Physical activity has been proven to have several protective effects which may prolong lifetime and decrease the incidence of several diseases, thus having a significant epidemiological impact on the society [1]. Many of these effects depend on the type, frequency, intensity and length of physical activity.

Several studies have confirmed the neuroprotective effect of physical activity on the central nervous system in the case of neurodegenerative diseases (such as Parkinson's disease, amyotrophic lateral sclerosis or schizophrenia) both in animal models and humans [2–7].

The protective effect of training on the retina has also been studied in animal models of retinal diseases. In mice, mild training on a treadmill has been confirmed to reduce light-induced retinal degeneration [8]. In a mouse model of glaucoma the protective effect of swimming on ganglion cell apoptosis has been described [9]. In streptozotocin-induced diabetic rats the apoptosis of the inner nuclear layer was blocked by treadmill training [10].

The acute changes induced by physical activity in the human body include the increase in blood pressure and heart rate that resultantly leads to an improved blood and oxygen delivery to the musculoskeletal system [11, 12]. At the same time, physical exercise may induce capillary constriction in the rest of the body through stress hormones that further helps the redistribution of circulation [13]. The retina is known to have an autoregulation similar to the brain which helps to maintain sufficient blood supply [14]. According to recent studies smaller retinal vessels are suggested to be significantly involved in the regulation of retinal blood flow and already minor changes in the arterial blood pressure can modify the retinal rheology [15–17]. This alteration may show differences in the macula and at the periphery [15].

Choroidal vasculature mainly consists of blood vessels with an intensive flow that provides nutrition for the outer retina and the cooling of the photoreceptors producing excessive heat during light signal processing [18]. According to studies on healthy volunteers, moderate physical activity leads to an increased blood flow in the choroid but not in the retina due to

retinal autoregulation [19–23]; however, there is also some evidence suggesting the presence of choroidal autoregulation, as well [14, 24].

Optical coherence tomography (OCT) provides a non-contact, non-invasive method for the in vivo, detailed examination of the retina. Custom built algorithms enable the segmentation of the OCT images and thus the measurement of the thickness of single intraretinal layers beyond total retinal thickness [25, 26]. OCT also enables the quantitative imaging of the choroidal vasculature by the option of enhanced depth imaging [27–31].

In the present pilot study, we aimed to assess the acute chorioretinal morphological changes in young sportsmen following short intense physical exercise using non-invasive spectral-domain optical coherence tomography (SD-OCT) imaging.

## Materials and methods

The study has been approved by the Semmelweis University Regional and Institutional Committee of Science and Research Ethics (272/2013) and written consent was obtained from all subjects in accordance with the Declaration of Helsinki.

Twenty-one left eyes of 21 healthy, young adults between 18–35 years of age were enrolled in this prospective study. Fifteen of them were professional athletes of the Hungarian Rowing Federation, whereas 6 subjects were healthy adults doing regular intensive physical activity (described as an increase in heart rate to vita maxima) at least 2 times a week. A survey questionnaire for general and ophthalmic history, the type and regularity of sports activity was completed by our subjects along with a question on any previous visual symptoms during heavy physical strain. Anthropometric variables such as height and weight were also recorded. Body mass index (BMI) was calculated as the ratio of weight to height in meter squared ($kg/m^2$).

Autorefractometry, uncorrected and best corrected visual acuity (measured on the ETDRS chart) and anterior and posterior segment examination with slit lamp was performed. Tropicamide (5 mg/ml) was used to dilate the pupil which was followed by a baseline OCT examination. Volumetric OCT scans of the macula were carried out by a Spectralis SD-OCT device (Heidelberg Engineering, Heidelberg, Germany) using the "Enhanced Depth Imaging" protocol to optimally visualize both retinal and choroidal structures. The option of Posterior Pole imaging with a setting of 30˚ (horizontally) × 25˚ (vertically) with Automatic Real Time (ART) of 20 was applied, containing 61 scans. Only scans with a signal strength value of at least 30 were accepted for the study.

The main inclusion criteria for the participants were the history of no ophthalmic or systemic diseases, ocular injury or operation, normal appearance of the macula when examined with biomicroscopy and best corrected visual acuity of at least 20/25 examined on the ETDRS vision chart. Subjects with a refractive error over ± 3D spherical equivalent were excluded from the study. A flowchart describing the recruitment process and the number of subjects included and excluded is shown on Fig 1.

None of the subjects drank alcohol or caffeine at least 24 hours before the exercise. The meals were not prescribed; however, all subjects had had their regular breakfast at least one hour before the exercise.

Each participant performed a stepwise incremental exercise trial until exhaustion (vita maxima) on a rowing ergometer (Concept 2 Type D, Morrisville, VT, USA). The intensity was increased every 500 meters so that the next load step had to be performed 10 seconds faster for the subjects who were rowing to complete fatigue. The maximum power achieved in the last load step was considered in the evaluation. The performance of each subject was expressed as the power-to-weight ratio (PWR, given in watt/kg) that allows a relatively neutral comparison between subjects [32, 33].

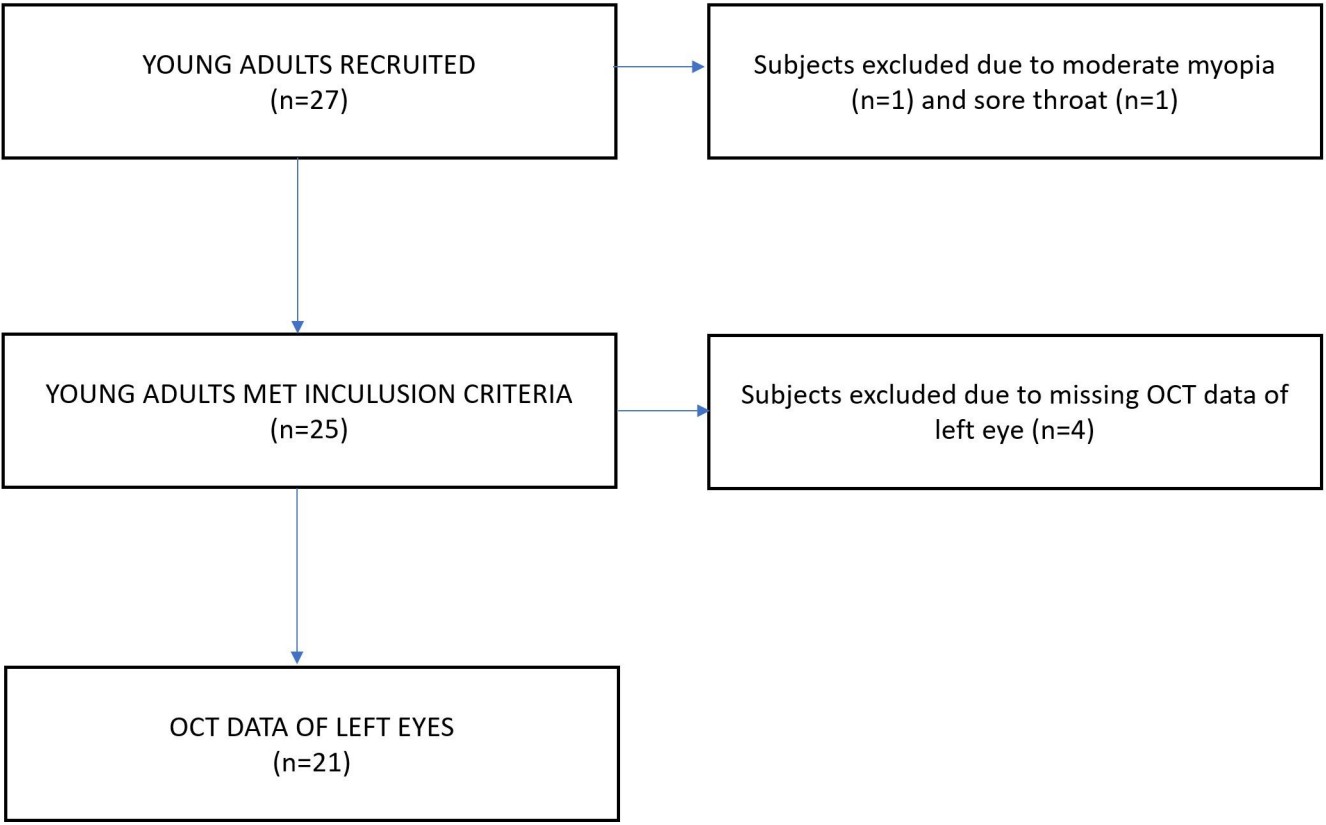

**Fig 1. Flowchart highlighting the recruitment process and the number of individuals included/excluded in the study.**

During the whole exercise and the recovery period, heart rate was monitored by a Polar Rs400® monitor (Polar Electro Oy, Kempele, Finland) in order to avoid reaching the maximum physiological age-related heart rate (calculated as 220 / min—age) [34]. Blood pressure (BP) was monitored by an Omron M6 Comfort® automatic cuff sphygmomanometer (Omron Healthcare Co. Ltd., Kyoto, Japan) before and 5 minutes after the exercise test. At the end of the post-exercise period the subjects were surveyed on visual symptoms during the rowing exercise.

OCT imaging was performed 1, 5, 15, 30, and 60 minutes following the rowing exercise. For each test, the eye tracker function was used to provide identical imaging of the retina during the course of the study. The baseline scans were set as reference and subsequent mapping took place at identical points.

The raw OCT data were exported from the OCT device and processed using our custom-built semiautomatic software (OCTRIMA 3D) described in detail previously [26]. The software runs on a MATLAB platform (The MathWorks Inc., Natick, MA, USA), collecting thickness data of the total macular volume and 7 retinal layers from the volumetric mapping of the macula along with the thickness of the choroid according to their reflectivity. The software allows semi-automatic image processing, i.e. the automatic designation of the boundaries of the layers are corrected manually during the review of the segmentation result. Previously, our group confirmed the high reproducibility of the OCTRIMA 3D segmentation of macular OCT scans in healthy subjects [35].

The thickness data of the total retina and of the following layers in the nine ETDRS regions (i.e., in the central subfield and in the superior, nasal, inferior, temporal regions in the inner

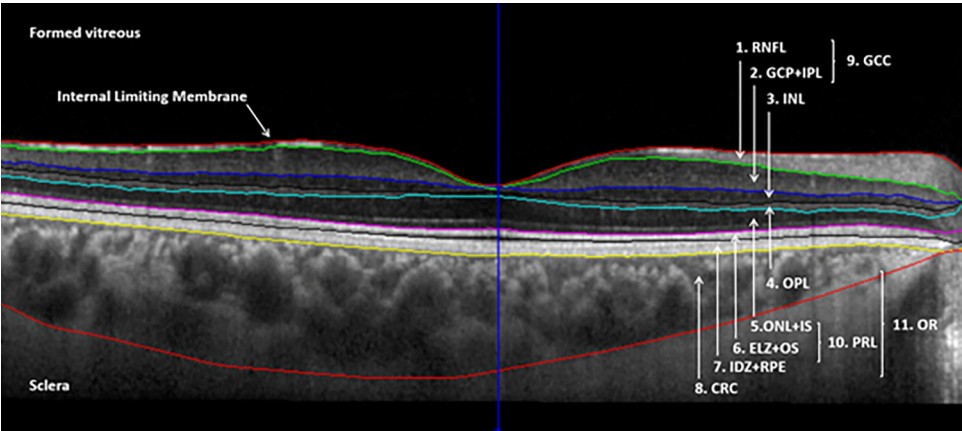

**Fig 2. Segmented macular OCT image showing all segmented boundaries by the OCTRIMA 3D algorithm.** The segmented retinal layers shown on the image are the following: retinal nerve fiber layer (RNFL), ganglion cell and inner plexiform layer complex (GCL+IPL), inner nuclear layer (INL), outer plexiform layer (OPL), complex layer containing the Henle fiber layer, outer nuclear layer, external limiting membrane and the myoid zone of the photoreceptors (ONL+IS), complex layer containing the ellipsoid zone and the outer segment of the photoreceptors (ELZ+OS), complex layer containing the interdigitation zone, retinal pigment epithelium and Bruch's complex (IDZ +RPE) and choroid containing the choriocapillaris, Sattler's layer and Haller's layer as far as the choroidal-scleral juncture (CRC). Composite layers, such as the ganglion cell complex (GCC), photoreceptor layer (PRL) and outer retina (OR) are also shown in the figure.

and the outer rings as well) were recorded: the retinal nerve fiber layer (RNFL), the complex layer of the ganglion cell and inner plexiform layer (GCL+IPL), the inner nuclear layer (INL), the outer plexiform layer (OPL), the complex layer containing the Henle fibers, outer nuclear layer, external limiting membrane and the myoid zone of the photoreceptors (ONL+IS), the complex layer of the ellipsoid zone and the outer segment of the photoreceptors (ELZ+OS), the complex layer containing the interdigitation zone, the retinal pigment epithelium and Bruch's membrane (IDZ+RPE) and the choroid (CRC) consisting of the choriocapillaris, Sattler's layer and Haller's layer as far as the choroidal-scleral juncture. (Fig 2) The above nomenclature follows the recommendation of the International Nomenclature for Optical Coherence Tomography Panel [36].

Beyond the single layers, composite layers were also created of anatomical and physiological consideration, such as the ganglion cell complex (GCC, RNFL+GCL+IPL), a complex containing the cellular elements of the photoreceptor layer (PRL, ONL+IS+ELZ+OS) and a complex of the outer retina (OR, OPL+ONL+IS+ELZ+OS+IDZ+RPE).

All OCT segmentation tasks were performed by the same experienced graders (ISZ, CSA and PF), supervised by a fourth experienced grader (GMS) who decided in the case of uncertainty. After the image processing step the thickness data of the retinal layers and choroid were recorded in four regions: for the total macula (T), the central subfield (1 mm in diameter, C) and the inner (I) and outer (O) macular rings (with diameters of 3 and 6 mm, I and O, respectively) (Fig 3).

Statistical analyses were carried out by SPSS Statistics for Windows, version 17.0 software (SPSS Inc., Chicago, Ill., USA). The Shapiro-Wilk test was used for normality testing. For normally distributed variables parametric tests were used and continuous data are reported as mean and standard deviation. The change of layer thickness from the baseline was calculated for each time point and one-way ANOVA test was performed for all variables, followed by Dunnett's post hoc test for the pairwise comparison between the thickness data at different time points and the baseline measurements. Pearson correlation was calculated to assess the

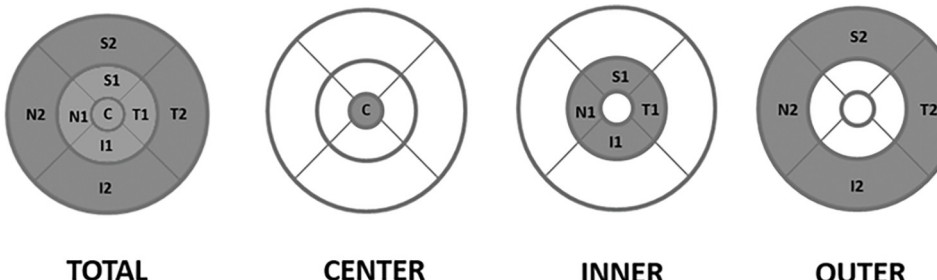

**Fig 3. The four macular regions in which the layer thickness was evaluated.** The total macula, the central subfield (1 mm in diameter) and the inner and outer macular rings (with diameters of 1, 3, and 6 mm, respectively).

correlation of retinal thickness changes and multiple linear regression models with stepwise method were executed in order to assess which layers are changing together. The Pearson correlation for performance and layer thickness changes at the 1st and 5th minute was also calculated to assess the effect of strain intensity and retinal changes. A subgroup analysis was performed to compare the baseline characteristics of professional vs. amateur sportsmen using Student's t-test and to assess potential differences in layer thickness change characteristics keeping age, gender, BMI under control. Due to the high number of comparisons the level of significance was set at 0.001; however, results with a *p value* between 0.001 and 0.05 are interpreted as missed significance.

## Results

Each participant did regular intensive physical activity. Professional sportsmen (n = 15) were rowers, while non-professional sportsmen (n = 6) did intensive workout at least twice a week. None of the subjects were smokers. The demographic data of the study subjects are shown in Table 1, while the detailed description of the participants is presented in S1 Table. More than one third (n = 8) of the participants confirmed having experienced eye symptoms during or after sports activities, most of them (n = 7) being professional sportsmen (for a detailed description of the symptoms see S1 Table). The mean age of the participants was 22.5 ± 4.1 years, the professionals were younger than amateurs (20.9 ± 2.5 and 26.5 ± 4.8 years, respectively, $p = 0.002$). The mean height was 1.78 ± 0.08 m and amateurs were shorter than professional sportsmen (1.70 ± 0.04 vs. 1.81 ± 0.06 cm, respectively, $p = 0.001$). For detailed descriptive statistics of the participants see Table 1 and S2 Table).

**Table 1. Descriptive statistics of the study participants.**

| | |
|---|---|
| **Age (years)** | 22.5 (4.1) |
| **Gender (m/f)** | 15/6 |
| **Height (m)** | 1.8 (0.1) |
| **Weight (kg)** | 72.7 (11.5) |
| **BMI (kg/m2)** | 22.8 (2.4) |
| **SBP (mmHg)** | 129.6 (14.1) |
| **DBP (mmHg)** | 74.3 (7.3) |
| **HR (1/min)** | 70.5 (13.1) |
| **SE (D)** | -0.2 (0.6) |

Abbreviations: Body mass index (BMI), systolic and diastolic blood pressure (SBP and DBP, respectively), heart rate (HR), spherical equivalent (SE). Data are presented as means (SD) (n = 21).

**Table 2. Layer thickness data of the study participants for total thickness of the entire macula (T), in the central subfield (C), inner ring (I) and outer ring (O).**

| Layers | T | C | I | O |
|---|---|---|---|---|
| RNFL | 38.2 (2.9) | 14.6 (1.5) | 26.7 (2.0) | 41.5 (3.3) |
| GCL+IPL | 77.0 (4.2) | 37.6 (7.3) | 100.3 (5.7) | 71.6 (4.2) |
| INL | 34.5 (1.8) | 19.7 (3.6) | 41.4 (2.6) | 33.0 (1.8) |
| OPL | 24.9 (1.8) | 18.7 (3.8) | 26.6 (2.2) | 24.6 (1.9) |
| ONL+IS | 79.5 (4.9) | 118.1 (9.2) | 92.5 (6.0) | 74.3 (4.8) |
| ELZ+OS | 31.8 (5.8) | 32.1 (3.8) | 31.0 (5.4) | 32.0 (6.0) |
| IDZ+RPE | 36.7 (5.8) | 43.9 (4.0) | 38.9 (5.1) | 35.7 (6.2) |
| CRC | 342.6 (68.0) | 379.5 (79.2) | 366.1 (76.4) | 334.3 (65.5) |
| **Composite layers** | | | | |
| GCC | 115.3 (6.4) | 52.3 (8.0) | 127.0 (6.9) | 113.1 (6.7) |
| PRL | 111.3 (6.9) | 150.3 (11.5) | 131.4 (8.5) | 110.0 (9.0) |
| OR | 172.8 (5.9) | 212.8 (9.6) | 189.1 (7.4) | 208.8 (12.9) |
| TR | 321.7 (8.8) | 284.7 (15.1) | 357.5 (11.2) | 312.7 (8.6) |

The data are shown as means (SD). For the abbreviations, see Fig 2.

The baseline thickness data of the different retinal layers are shown in Table 2. For the layer thickness differences between the professional and amateur sportsmen see S3 Table.

For the entire cohort, BCVA and gender had no effect on baseline thickness data. The baseline INL in the outer ring and the entire macula showed positive correlation with weight. Positive correlation was found between height and the baseline ONL+IS in the center. Diastolic blood pressure negatively correlated with baseline ELZ+OS in the inner ring and the entire macula. The above correlations and correlations with missed significance (with a p value between 0.001 and 0.05) are also interpreted in Table 3. Age seemed to have a negative correlation with the thickness of the outer retinal layers and the choroid, whereas data were suggestive of a positive correlation between height and choroidal thickness along with diastolic blood pressure and thickness of the outer retinal layers.

We observed a significant thinning of the total retina 1 minute post-exercise (-7.3 ± 0.6 μm, $p < 0.001$) which was followed by a significant thickening at 5 and 15 minutes (+3.6 ± 0.6 μm and +4.0 ± 0.6 μm, respectively, for both $p < 0.001$). By 30 minutes total retinal thickness returned to baseline. These changes were significant in the inner and outer ring as well, but not in the central subfield (Fig 4).

This trend above was present throughout the most layers of the retina, with significant changes in the GCL+IPL layer complex at 1, 5 and 15 minutes (-1.3 ± 0.1 μm, +0.6 ± 0.1 μm and +0.7 ± 0.1 μm, respectively, $p < 0.001$ for all), in the INL at 1 and 5 minutes (-0.8 ± 0.1 μm and +0.8 ± 0.1 μm, respectively, $p < 0.001$ for both), in the ONL+IS at 1, 5, 15 and 60 minutes (-1.4 ± 0.4 μm, $p = 0.003$; +2.3 ± 0.4 μm, $p < 0.001$; +1.1 ± 0.1 μm, $p = 0.031$; and -1.1 ± 0.4 μm $p = 0.044$, respectively) and in the IDZ+RPE complex at 1 and 15 minutes (-3.3 ± 0.4 μm and +1.8 ± 0.4 μm, respectively, $p < 0.001$ for both) (Fig 5).

We assessed the physiologically different parts of the retina as composite layers, as well (GCC, PRL and OR). The GCC thickness changes were significant for the total retinal thickness, in the inner and outer rings at 1 minute and 15 minutes. In the case of the PRL and OR, there were significant changes for the total retinal thickness and all three macular regions, as well (Fig 6, data are shown in S5 Table).

There was no significant change observed in choroidal thickness; however, we could detect a tendency towards thinning at 1, 15 and 30 minutes following exercise (Fig 7) The absolute

**Table 3. Correlations between demographic characteristics and baseline layer thickness values.**

| Layers | Age | | SE | | Height | | Weight | | BMI | | DBP | | HR | |
|---|---|---|---|---|---|---|---|---|---|---|---|---|---|---|
| | R | p | R | p | R | p | R | p | R | p | R | p | R | p |
| RNFL_T | -0.299 | 0.187 | 0.052 | 0.821 | 0.026 | 0.912 | -0.147 | 0.526 | -0.224 | 0.328 | -0.015 | 0.950 | -0.450 | 0.040 # |
| RNFL_C | -0.507 | 0.019 # | 0.302 | 0.184 | 0.247 | 0.281 | -0.014 | 0.953 | -0.216 | 0.346 | -0.309 | 0.173 | -0.424 | 0.055 |
| RNFL_I | -0.310 | 0.172 | 0.127 | 0.582 | 0.159 | 0.491 | -0.156 | 0.500 | -0.339 | 0.133 | -0.116 | 0.616 | -0.484 | 0.026 # |
| GCL+IPL_I | -0.338 | 0.134 | 0.387 | 0.083 | 0.467 | 0.033 # | 0.335 | 0.137 | 0.131 | 0.571 | -0.017 | 0.943 | -0.186 | 0.420 |
| INL_T | -0.222 | 0.334 | 0.275 | 0.228 | 0.542 | 0.011 # | 0.665 | **0.001** | 0.535 | 0.012 # | -0.095 | 0.684 | -0.103 | 0.658 |
| INL_C | 0.286 | 0.209 | 0.139 | 0.548 | -0.123 | 0.595 | -0.007 | 0.977 | 0.127 | 0.584 | 0.450 | 0.041 | 0.174 | 0.450 |
| INL_I | -0.174 | 0.451 | 0.397 | 0.075 | 0.464 | 0.034 # | 0.525 | 0.015 # | 0.405 | 0.068 | 0.104 | 0.655 | -0.051 | 0.826 |
| INL_O | -0.241 | 0.294 | 0.187 | 0.418 | 0.530 | 0.014 # | 0.658 | **0.001** | 0.527 | 0.014 # | -0.200 | 0.385 | -0.126 | 0.585 |
| ONL+IS_C | -0.374 | 0.095 | 0.085 | 0.715 | 0.654 | **0.001** | 0.305 | 0.179 | -0.094 | 0.684 | -0.390 | 0.081 | -0.326 | 0.149 |
| ELZ+OS_T | -0.565 | 0.008 # | 0.202 | 0.379 | 0.124 | 0.592 | 0.212 | 0.356 | 0.192 | 0.406 | -0.652 | **0.001** | -0.244 | 0.287 |
| ELZ+OS_C | -0.632 | 0.002 # | 0.184 | 0.425 | 0.271 | 0.235 | 0.088 | 0.704 | -0.103 | 0.657 | -0.545 | 0.011 # | -0.453 | 0.039 # |
| ELZ+OS_I | -0.626 | 0.002 # | 0.233 | 0.308 | 0.186 | 0.420 | 0.213 | 0.355 | 0.144 | 0.533 | -0.664 | **0.001** | -0.271 | 0.234 |
| ELZ+OS_O | -0.541 | 0.011 # | 0.192 | 0.404 | 0.103 | 0.656 | 0.213 | 0.355 | 0.209 | 0.364 | -0.644 | 0.002 # | -0.229 | 0.318 |
| IDZ+RPE_T | 0.476 | 0.029 # | -0.174 | 0.451 | -0.119 | 0.606 | -0.251 | 0.273 | -0.250 | 0.273 | 0.543 | 0.011 # | 0.393 | 0.078 |
| IDZ+RPE_C | 0.512 | 0.018 # | -0.244 | 0.287 | -0.410 | 0.065 | -0.236 | 0.304 | 0.000 | 0.999 | 0.352 | 0.117 | 0.535 | 0.012 # |
| IDZ+RPE_I | 0.548 | 0.010 # | -0.199 | 0.388 | -0.247 | 0.281 | -0.284 | 0.213 | -0.198 | 0.391 | 0.567 | 0.007 # | 0.475 | 0.030 # |
| IDZ+RPE_O | 0.450 | 0.041 # | -0.163 | 0.480 | -0.080 | 0.732 | -0.239 | 0.298 | -0.265 | 0.246 | 0.532 | 0.013 # | 0.362 | 0.106 |
| PRL_T | -0.616 | 0.003 # | 0.036 | 0.877 | 0.261 | 0.253 | 0.170 | 0.461 | 0.015 | 0.949 | -0.622 | 0.003 # | -0.324 | 0.152 |
| PRL_C | -0.513 | 0.018 # | 0.130 | 0.575 | 0.616 | 0.003 # | 0.275 | 0.228 | -0.110 | 0.634 | -0.496 | 0.022 # | -0.414 | 0.062 |
| OR_C | -0.445 | 0.043 # | 0.110 | 0.635 | 0.451 | 0.040 # | 0.179 | 0.436 | -0.104 | 0.652 | -0.401 | 0.072 | -0.251 | 0.273 |
| CRC_T | -0.542 | 0.011 # | 0.504 | 0.020 # | 0.514 | 0.017 # | 0.128 | 0.582 | -0.216 | 0.348 | -0.341 | 0.130 | -0.031 | 0.895 |
| CRC_C | -0.474 | 0.030 # | 0.426 | 0.054 | 0.577 | 0.006 # | 0.156 | 0.500 | -0.234 | 0.307 | -0.295 | 0.195 | -0.146 | 0.526 |
| CRC_I | -0.519 | 0.016 # | 0.447 | 0.042 # | 0.531 | 0.013 # | 0.114 | 0.622 | -0.255 | 0.265 | -0.332 | 0.142 | -0.116 | 0.617 |
| CRC_O | -0.550 | 0.010 # | 0.524 | 0.015 # | 0.503 | 0.020 # | 0.130 | 0.574 | -0.200 | 0.385 | -0.344 | 0.126 | 0.004 | 0.986 |

The table shows the Pearson correlation coefficients with the corresponding *p* values. Significant data are highlighted in bold, # denotes missed significant results (with p values between 0.001 and 0.05). Only layers with any significant or missed significant correlations are shown. (For the abbreviations of the layers see Fig 2).

changes in choroidal thickness did not show any correlation with the thickness changes of the intraretinal layers.

Multiple linear regression did not reveal any significant confounders. Keeping the confounding factors gender, height, weight, BMI, SBP, fitness level under control, no significant difference was shown for the comparison of amateur vs. professional sportsmen.

However, significant correlation between PWR and any layer thickness changes was found only in the case of INL in the central subfield after 5 minutes. The correlation coefficients along with the "missed significant" *p* values are also shown in Table 4. There was a statistically

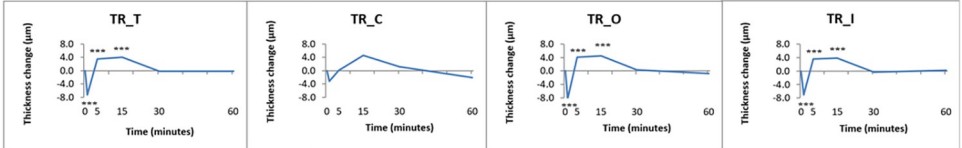

**Fig 4. Changes of total retinal thickness over time following vita maxima strain of the study participants.** Data are shown in the total macular area (T), the central subfield (C), the inner (I) and outer ring (O). *: *p* <0.05 (missed significance), **: *p* <0.01 (missed significance), ***: *p* <0.001 (significant). For mean and SD values see S4 Table. (For the abbreviations of the layers see Fig 2).

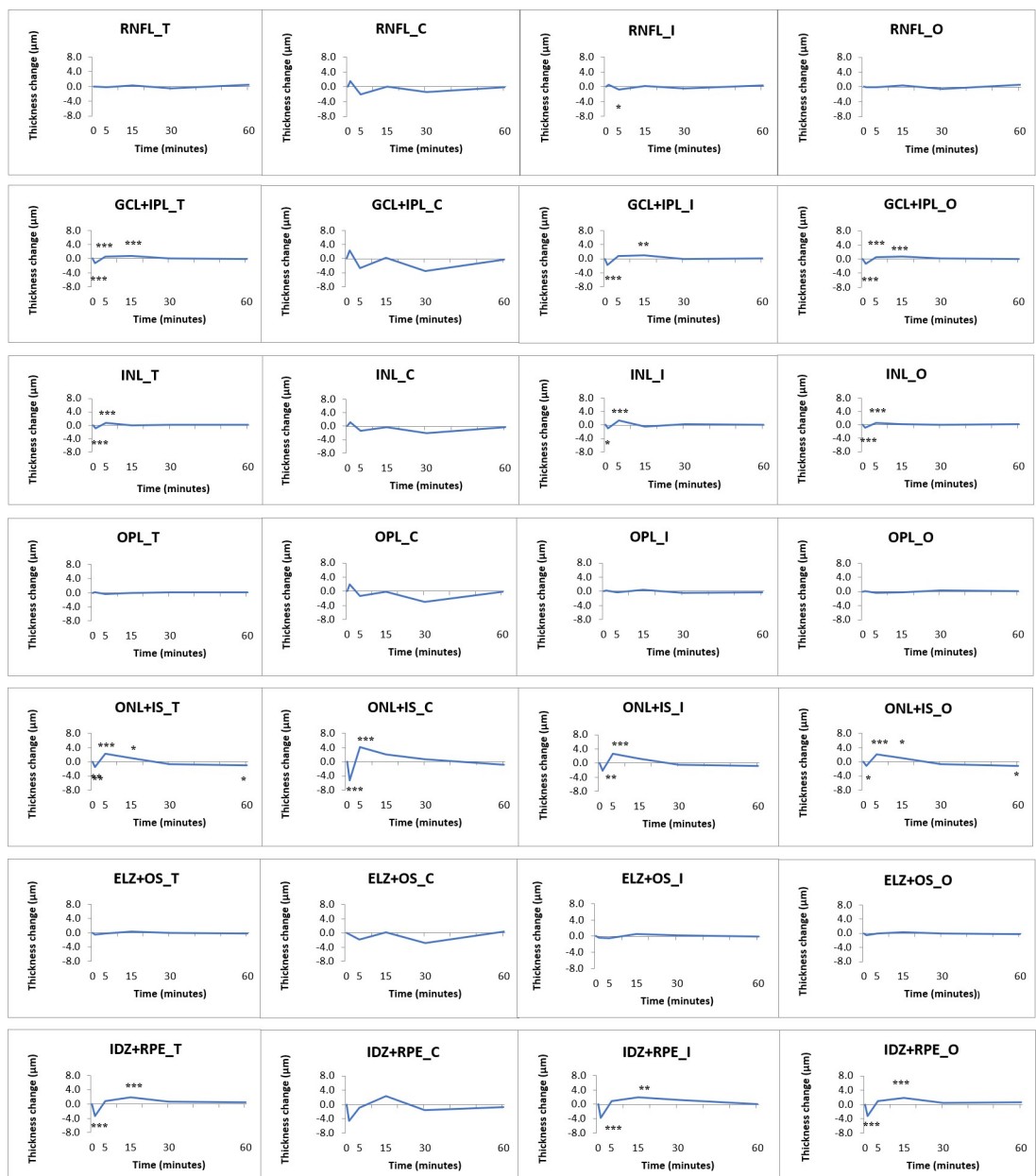

**Fig 5. Changes of single retinal layer thickness over time following vita maxima strain of the study participants.** Data are shown in the total macular area (T), the central subfield (C), the inner (I) and outer ring (O). *: $p < 0.05$ (missed significance), **: $p < 0.01$ (missed significance), ***: $p < 0.001$ (significant). For mean and SD values see S4 Table. (For the abbreviations of the layers see Fig 2).

significant difference in PWR between athletes and amateurs (5.5 ± 0.6 vs. 3.4 ± 0.8 watt/kg, respectively, $p < 0.001$).

## Discussion

There is only relatively few evidence available on the direct effects of physical exercise on the retina and choroid. In this study, rather pilot in nature due to the sample size, the morphological effects of short intense physical activity on the retina and choroid were examined in vivo in

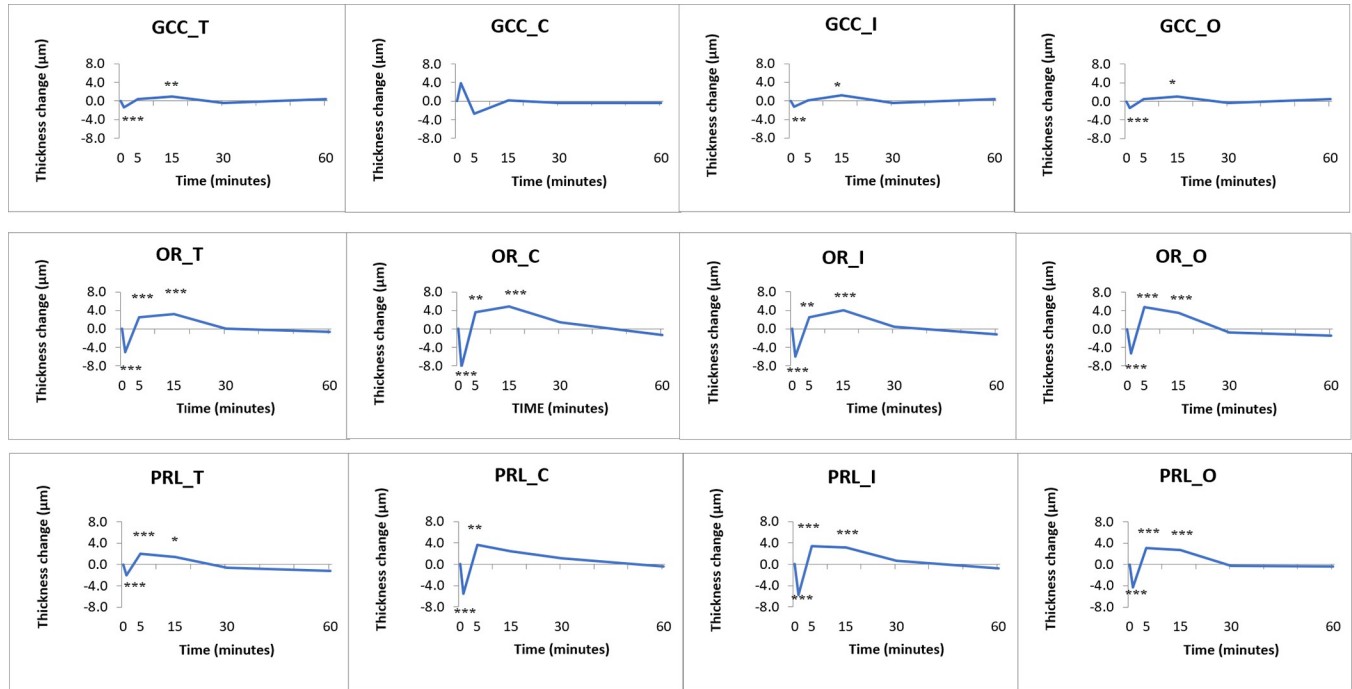

**Fig 6. Changes observed in the composite layers of the macula.** Data are shown in the total macular area (T), the central subfield (C), the inner (I) and outer ring (O). From top to bottom: GCC (RNFL+GCL+IPL), OR (OPL+ONL+IS+ELZ+OS+IDZ+RPE) and PRL (ONL+IS+ELZ+OS). *: $p <0.05$ (missed significance), **: $p <0.01$ (missed significance), ***: $p <0.001$ (significant). For the mean and SD values see S5 Table. (For the abbreviations of the layers see Fig 2).

young, physically active adults. To detect the structural alterations on the optical coherence tomographic images, the OCTRIMA 3D software developed by our research team was used. We found that there is an acute thinning at one minute, followed by an immediate refractory thickening of the retina by five minutes post-exercise that lasts until the complete restitution at 30 minutes. This acute change is most pronouncedly present in the granular layers of the retina and is not related to the performance delivered during exercise and seems to be independent of professional or amateur status.

The reason behind our observations is unclear and remains to be elucidated. We speculate at least two mechanisms to be involved. First, it is known that a systolic blood pressure rise exceeding 20 mmHg triggers vasoconstriction in the retinal arteriolar system [15, 16] which could explain the acute thinning of the GCL+IPL and INL layers which are known to contain cell bodies and contain the superficial and deep capillary plexi. Second, by the 5th post-exercise minute, the restitution begins, the blood pressure decreases and vasoconstriction is not any more dominant, resulting in refractory thickening of the above granular layers.

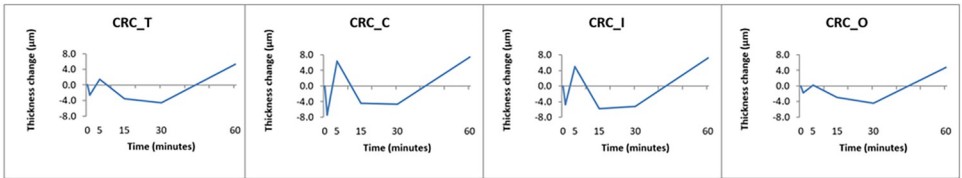

**Fig 7. Changes observed in the choroid over time following vita maxima strain of the study participants.** The changes were not statistically significant (from left to right, macular area (T), central subfield (C), inner (I) and outer rings(O)). For mean and SD values see S4 Table. (For the abbreviations of the layers see Fig 2).

**Table 4. Correlation between the performance (maximal power-to-weight ratio expressed as watt/kg) and the layer thickness changes at 1 and 5 minutes of the recovery period for the total retina and three macular areas (T, total macula, C, central subfield, I, inner ring, O, outer ring).**

| 1 min | GCL+IPL_I | INL_ T | INL_I | CRC_C | CRC_O | |
|---|---|---|---|---|---|---|
| PWR | -0.44 | 0.502 | 0.553 | 0.472 | 0.501 | |
| *p* | 0.046 # | 0.021 # | 0.009 # | 0.031 # | 0.021 # | |
| 5 min | GCL+IPL_T | GCL+IPL_I | GCL+IPL_O | INL_C | OPL_T | CRC_O |
| PWR | 0.584 | 0.542 | 0.481 | -0.668 | -0.44 | -0.503 |
| *p* | 0.005 # | 0.011 # | 0.027 # | **0.001** | 0.046 # | 0.020 # |

Only significant and missed significant Pearson correlations observed at 1 and 5 minutes post-exercise are shown (highlighted in bold and denoted with #, respectively).

The main mechanism underlying retinal autoregulation is the myogenic response of the blood vessel wall to blood pressure fluctuations, i.e. vasoconstriction occurs in small arteries and arterioles with an increase in transmural pressure; as the internal pressure decreases, the blood vessels dilate which is in line with our observations, as well [16]. This blood-pressure driven change has been shown in vivo in the retina [15–17], where a systolic rise of 15–20 mmHg led to a reactive vasoconstriction. It is assumed that this response is mainly driven by endothelial cells and other local mediators (oxygen, carbon dioxide, angiotensin-II, adenosine, nitric oxide, endothelin-1) [37].

Since the outer retina contains no vascular elements, the explanation for the acute thinning and then rebound thickening of the ONL+IS and IDZ+RPE layers must be either a metabolic or a biomechanic change. It is known that acute physical strain leads to an increase in intraocular pressure (IOP). This pressure rise can lead to a mechanical compression of the entire retina, including the photoreceptors and RPE cells. This speculation of mechanical compressions seems feasible, on the one hand, due to the fact that the nuclear layers, containing the cell nuclei of the retina, showed changes whereas the axonal layers remained mostly intact. It can be speculated that the nuclei allow more volume changes than the fibrillar layers (such as the RNFL, IPL and OPL). As hyperbaric changes have recently been described to result in acute shortening of the photoreceptors, it is possible that metabolic factors may also play a role in the observed phenomena [38].

Our presumption was that changes in the choroid may drive the acute changes of the outer retina. However, we observed no clear trend in the choroid and correspondingly no correlation was found between retinal and choroidal thickness.

The trends observed in the choroid were somewhat surprising, with an acute thinning followed by immediate thickening and then further, sustained thinning. The reason for this is unclear as the investigation of choroidal changes due to physical exercise is rather controversial [23, 39–44].

In some studies, macular perfusion and retinal vascular density were reported to decrease significantly after exercise [39, 42] which may explain the acute thinning of the choroid just after the physical strain. In our study, after the trend for acute thinning in the first post-exercise minute, we observed a thickening trend at 5 minutes, followed by another thinning trend at 30 minutes and then a slow thickening again, which may refer to an extended choroidal reaction and regeneration. Li et al. also reported a decrease in choroidal thickness after moderate physical activity that lasted for at least 30 minutes; however, they did not examine the first 10 minutes of the recovery period [42]. The dilatation or contraction of the vascular or non-vascular smooth muscles of the choroid may modulate the blood flow [41]. Thus, the autoregulation of the choroid may act through the sympathetic innervation of the capillary system in response to an increase or decrease in IOP [18]. However, choroidal autoregulation is being

the subject of debate. According to some studies, choroidal circulation does not change in healthy eyes even in the case of sudden elevation of the perfusion pressure by 97.5%, while in patients with age-related macular degeneration only 23% elevation of the perfusion pressure increased choroidal flow [43]. In another setting, baseline choroidal blood flow measured by confocal laser Doppler flowmeter before and after a physical load has been shown to be lower in healthy subjects, compared to those who suffered from open angle glaucoma. After exercise, the perfusion increase was twice as high in case of glaucoma as in the healthy group [45].

The existence of choroidal autoregulation is also supported by another research, which reported that choroidal blood flow was slightly dependent on mean arterial pressure when the exercise-induced MAP elevation was smaller than 25mmHg, but over 25mmHg, MAP had no effect [46]. Choroidal alterations on SD-OCT after physical exercise were studied in healthy subjects where only systolic blood pressure increased, choroidal thickness remained constant [40, 47].

In contrast, Sayin et al have shown that in young healthy male (ranged in age from 23 to 33 years old) subjects the thickness of the choroid increased in the first 5 minutes after 10 minutes of low-efficiency, medium intensity training and then returned to the baseline at 15 minutes after the exercise, while, interestingly, retinal thickness did not change [23]. We believe the difference to our study was most possibly due to different exercise intensity as in our study we strained our subject till complete fatigue. As mentioned above, we suspected a correlation between performance and retinal or choroidal changes which we could not confirm [40, 48].

The performance of our study subjects showed some variation as, not surprisingly, professional sportsmen delivered higher workload. However, no real correlations were found with chorioretinal changes except for the GCL+IPL layer at 5 minutes post-activity. As the GCL+IPL contains the inner plexus of the retinal capillaries, this might point towards the involvement of acute stress in the changes of the inner retina; another mechanism could be in part the mechanical compression of this relatively thick layer at the perifoveal retina abundant in ganglion cell bodies.

In subjects undergoing high intensity interval training (HIIT) for 4 weeks, an inverse correlation between individual fitness and FAZ area was described, whereas no further correlations between other OCTA parameters and individual physical parameters were found. In response to HIIT during the study, the mean FAZ area in the deep retinal plexus and macular flow density of the superficial layer significantly decreased while flow density in the peripapillary area showed an increase [49]. In contrast to this, in type 1 diabetic patients following a 4-week HIIT exercise none of the analyzed microvascular parameters changed in response to the intervention [50].

Another study in cyclists described an increase in critical flicker frequency, a parameter indirectly indicative of optic nerve function. In this cohort, CFF increased immediately after training that was maintained 30 min post exercise which is in accordance with our observations regarding physiological restitution of the posterior pole after 15 minutes following high physical strain [51].

Although there is little known about the acute biochemical effects of physical exercise in humans, there are some evidences available from animal studies. As mentioned above, in streptozotocin induced diabetic rats, the apoptosis induced by diabetes decreased in the retinal cells due to the elevated p-Akt expression caused by treadmill training [9]. In another animal study, following toxic light impulse (10000 lux) twice as many photoreceptors became damaged in non-trained mice as in the trained group. The BDNF (brain-derived neurotrophic factor) level was also higher in the trained group, whereas the protective effect of physical exercise was blocked by the neutralization of the BDNF effect by a systemic receptor antagonist (A12), suggesting an important role of BDNF in physical activity-related neuroprotection [8].

It is important to note that our work has certain limitations. First, a relatively small number of cases was recruited, especially in relation of the professional vs. amateur comparison. Thus, in order to more accurately assess the correlation between retinal and choroidal parameters with age, SE, height, weight, BMI, DBP and HR a larger number of subjects would be required. However, our study was rather of a pilot nature on which future studies can be based and therefore we did not plan to include a larger cohort. For technical reasons axial length was not measured that might have influenced the segmentation results [52]. We believe this did not affect our results as we looked at layer thickness changes from baseline rather than absolute thickness in which way such a bias could be avoided. Intraocular pressure might have had an influence on our results (as mentioned above); however, we could not measure IOP after (and during) the physical load/strain as it would have been technically very demanding parallel with OCT imaging, the latter being of primary importance to us in this study. Besides, there are literature data already available on the acute IOP rise immediately after physical exercise [53]. Regarding sports activities (intensity and type of sport) our subjects have also shown variability, as well and there was no control group with sedentary lifestyle, either, that might have an effect on the maximum performance obtained on a rowing pad. We believe this variability was of less importance as we used a rather general output of watt/kg that eliminates this variability to some extent. Finally, we could not measure blood pressure values during exercise (due to the fact that we used a rowing ergometer), but on the other hand, the heart rates were continuously monitored and were used for the calculation of power. The power data showed a certain difference in the subjects' performance, but all subjects were strained until complete fatigue that led to a similar physiological response in all subjects. Besides, total workload/performance did not show a strong correlation with retinal changes. Interestingly, none of the participants reported a visual symptom on the post-exercise questionnaire, whereas in the pre-exercise questionnaire some professional sportsmen reported to have experienced such symptoms during heavy strain.

To our knowledge, our work is the first to describe the acute morphological changes of the retina and choroid due to heavy physical exercise in a young adult population. We could show the acute thinning followed by rebound thickening of the granular layers of the retina with complete restitution by 30 minutes post-exercise. These changes could serve as one possible explanation for the observation of blackout during heavy physical exercise described by professional sportsmen, although the exact reasons remain unclear. We hypothesize the combination of acute stress-related vascular changes of the inner retina combined with biomechanical changes of the outer retina due to IOP increase to be in the background of these changes. The choroid neither seem to play a role in these changes nor it shows any acute alterations due to heavy physical strain, suggesting physiological processes maintaining constant morphological parameters under such conditions. The relevance of our findings remains so far unclear; we believe that acute retinal effects of physical activity and regular exercise may play an important role in maintaining eye health and thus may chronically lead to an entity that could be described as the "trained eye" [54], similarly to that referred to as the "trained heart" in cardiology. However, our theory warrants further investigation in the future in a larger cohort and in a prospective manner, including subjects with retinal pathologies, as well.

## Supporting information

**S1 Table. Descriptive data of the study participants.** Amateur vs. professional status, type of sports, systemic and ocular symptoms experienced during and after sports activity are highlighted. Abbreviations: body mass index (BMI), heart rate (HR), systolic and diastolic blood pressure (SBP and DBP, respectively), power to weight ratio (PWR, in the case of Subject

4 the data were corrupted during processing).
(TIF)

**S2 Table. Descriptive statistics of the study participants.** Data are presented as means (SD), the *p* value refers to the comparison by Student's t-test between professional (n = 15) and amateur sportsmen (n = 6). Significant data are highlighted in bold, # denotes missed significant results (with p values between 0.001 and 0.05).
(TIF)

**S3 Table. Layer thickness differences for the professional and amateur sportsmen.** For the abbreviations see Fig 2.
(TIF)

**S4 Table. Thickness changes of the single retinal layers and the choroid in the study participants calculated for the entire macula, for the central subfield, the inner and outer rings of the macula.** The *p* values indicate the results of the post hoc Dunnett test in the case of significant ANOVA test.
(TIF)

**S5 Table. Changes of layer thickness data of the composite layers of the macula in the study participants calculated for the entire macula, for the central subfield, the inner and outer pericentral rings.** The composite layers are the following: ganglion cell complex (GCC, RNFL+GCL+IPL), outer retina (OR, OPL+ONL+IS+ELZ+IS+IDZ+RPE) and photoreceptor layer (PRL, ONL+IS+ELZ+OS). The *p* values indicate the results of the post hoc Dunnett test in the case of significant ANOVA test. (For the abbreviations see Fig 2).
(TIF)

## Acknowledgments

The authors are grateful to Eszter Szendrei for her valuable assistance in performing the physical strain of the subjects.

## Author Contributions

**Conceptualization:** Irén Szalai, Miklós Tóth, Gábor Márk Somfai.

**Data curation:** Anita Csorba, Endre Horváth, István Györe.

**Formal analysis:** Irén Szalai, Gábor Márk Somfai.

**Investigation:** Irén Szalai, Fanni Pálya, Edit Bosnyák.

**Methodology:** Endre Horváth, Edit Bosnyák, István Györe, Gábor Márk Somfai.

**Project administration:** Irén Szalai, Anita Csorba, Fanni Pálya.

**Resources:** Zoltán Zsolt Nagy, Gábor Márk Somfai.

**Software:** Tian Jing.

**Supervision:** Zoltán Zsolt Nagy, Delia Cabrera DeBuc, Miklós Tóth, Gábor Márk Somfai.

**Validation:** Tian Jing, Gábor Márk Somfai.

**Visualization:** Irén Szalai, Endre Horváth.

**Writing – original draft:** Irén Szalai.

**Writing – review & editing:** Gábor Márk Somfai.

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
