## [Decision Letter · Decision Letter 0]

9 Nov 2021

PONE-D-21-23138The assessment of acute chorioretinal changes due to intensive physical exercise in young adultsPLOS ONE

Dear Dr. Somfai,

Thank you for submitting your manuscript to PLOS ONE. After careful consideration, we feel that it has merit but does not fully meet PLOS ONE’s publication criteria as it currently stands. Therefore, we invite you to submit a revised version of the manuscript that addresses the points raised during the review process.

 Please submit your revised manuscript by Dec 24 2021 11:59PM. If you will need more time than this to complete your revisions, please reply to this message or contact the journal office at plosone@plos.org. Please include the following items when submitting your revised manuscript:A rebuttal letter that responds to each point raised by the academic editor and reviewer(s). You should upload this letter as a separate file labeled 'Response to Reviewers'.A marked-up copy of your manuscript that highlights changes made to the original version. You should upload this as a separate file labeled 'Revised Manuscript with Track Changes'.An unmarked version of your revised paper without tracked changes. You should upload this as a separate file labeled 'Manuscript'.

We look forward to receiving your revised manuscript.

Kind regards,

Andrzej Grzybowski

Academic Editor

PLOS ONE

Journal Requirements:

2. Please note that in order to use the direct billing option the corresponding author must be affiliated with the chosen institute. Please either amend your manuscript to change the affiliation or corresponding author, or email us at plosone@plos.org with a request to remove this option.

Reviewers' comments:

Reviewer's Responses to Questions

**Comments to the Author**

1. Is the manuscript technically sound, and do the data support the conclusions?

Reviewer #1: Partly

Reviewer #2: Yes

2. Has the statistical analysis been performed appropriately and rigorously? 

Reviewer #1: No

Reviewer #2: Yes

3. Have the authors made all data underlying the findings in their manuscript fully available?

Reviewer #1: Yes

Reviewer #2: Yes

4. Is the manuscript presented in an intelligible fashion and written in standard English?

Reviewer #1: No

Reviewer #2: Yes

5. Review Comments to the Author

Reviewer #1: First, i would like to congratulate the authors for their amazing work on investigating the correlation between chorioretinal thickness changes following intensive physical exercise.

Although the authors have fully and thoroughly highlighted the limitations to their research in the Discussion section, there are some concerns that should be addressed before this manuscript can be deemed significant for publication.

1- Using Pearson's correlation to examine the changes in chorioretinal thickness is a good choice given the lack of normality in the recorded observations. However, i think it would be better to conduct multiple linear regression models instead. This will help us determine if the change in chorioretinal thickness is really a reflection of change in physical activity that is not confounded by other baseline variables. The results of this analysis will not be definite as well, since certain confounders were not assessed during the recruitment period (i.e., axial length, IOP measurement), and this has to still be mentioned in the limitations section. I would suggest using time-dependent dependent variables. For example, the dependent variables would be: choroidal thickness at baseline; choroidal thickness change (from baseline to 5 minutes); choroidal thickness change (from 5 mins to 15 mins) and so on. This would give a better comprehension of the actual change in such measurement during each time interval.

2- Did you put a limitation of participants' age during recruitment? (for example, >18 years of age)

If yes, then it needs to be mentioned in the methods section. And, if not, then based on what did you categorize them as young age?

3- I would recommend adding a flow chart highlighting the recruitment process and the number of individuals included/excluded with reasons.

4- Please replace reference 29 with a more updated/recent one.

5- Please improve the quality of the figures provided.

6- In the introduction section: "The acute changes induced..............blood supply." Please add references that support each of the claims provided.

7- In the conclusion section, both in the main text and in the abstract, please restrict your conclusions to the young age population.

8- Please remove the company name of the SD-OCT device from the abstract section.

9- The English structure of the manuscript needs revision and editing by a native English speaker.

Reviewer #2: The authors describe a prospective study of the effect of exercise on the thickness of retino-choroidal layers in twenty-one eyes (four excluded due to poor data quality), in a population of young adults.

Retina-choroidal thickness was recorded in the 9 ETDRS regions, and was measured using OCT. Image analysis and segmentation was performed using custom software. Correlation was analysed between retina-choroidal thickness pre and post exercise. Other biometric parameters (height, weight, body mass index, blood pressure, heart rate, and refraction) were considered. The authors concluded that there was retinal thinning followed by thickening post exercise, no significant correlation between the retinal and choroidal changes were detected. Two mechanisms are discussed: circulatory auto-regulatory mechanisms and a mechanical effect of IOP on retinal thickness.

Comments:

The study was generally well designed, the statistical analysis was sound and the literature adequately reviewed. The omission of IOP measurement limits the interpretability of the findings particularly when the authors conclude that this may be one of the mechanisms that could explain the study findings and this factor is easily measured.

The article is technically sound, The statistical tests have been applied rigorously, the authors have made the data available for analysis and the manuscript is written in intelligible standard.

Minor recommendations:

Although the authors have used the standard conventions for abbreviations, the number of the abbreviations limits the readability, perhaps if a list of abbreviations could be provided in the end of the article.

6. PLOS authors have the option to publish the peer review history of their article (what does this mean?). If published, this will include your full peer review and any attached files.

Reviewer #1: **Yes: **Abdelaziz Abdelaal

Reviewer #2: No

---

## [Author Response · Author response to Decision Letter 0]

23 Dec 2021

We would like to thank both reviewers for their insightful comments. Please find below our detailed comments for all points raised in the review.

Reviewer #1: First, i would like to congratulate the authors for their amazing work on investigating the correlation between chorioretinal thickness changes following intensive physical exercise.

We would like to thank Reviewer #1 for his positive comment.

Although the authors have fully and thoroughly highlighted the limitations to their research in the Discussion section, there are some concerns that should be addressed before this manuscript can be deemed significant for publication.

1- Using Pearson's correlation to examine the changes in chorioretinal thickness is a good choice given the lack of normality in the recorded observations. However, i think it would be better to conduct multiple linear regression models instead. This will help us determine if the change in chorioretinal thickness is really a reflection of change in physical activity that is not confounded by other baseline variables. The results of this analysis will not be definite as well, since certain confounders were not assessed during the recruitment period (i.e., axial length, IOP measurement), and this has to still be mentioned in the limitations section. I would suggest using time-dependent dependent variables. For example, the dependent variables would be: choroidal thickness at baseline; choroidal thickness change (from baseline to 5 minutes); choroidal thickness change (from 5 mins to 15 mins) and so on. This would give a better comprehension of the actual change in such measurement during each time interval.

We agree with Reviewer #1 on the use of multiple linear regression and we indeed applied this methodology in our calculations. (See Page 10 Line 199: “Pearson correlation was calculated to assess the correlation of retinal thickness changes and multiple linear regression models with stepwise method were executed in order to assess which layers are changing together.”) As no results for confounders were found, we did not include anything in the text. To correct this, we now added a sentence in the results section referring to this. (See Page 16 Line 296: “Multiple linear regression did not reveal any significant confounders.”) In the limitations section we are mentioning the lack of axial length and IOP measurements as a potential limiting factor of our study. (See Page 22 Line 424)

We also agree on the use of time-dependent dependent variables between adjacent time points. We did consider this at the time of statistical design but decided to use the baseline measurements for comparisons, i.e. baseline to 1 minute, baseline to 5 minutes and so forth, in order to assess relative change due to physical exercise. This also enables us to show thickness values of the retinal layers at different time points (and not the change values) on our figures that allows for a more intuitive perception and interpretation of our data. For the above reasons, we would remain with our original methodology.

2- Did you put a limitation of participants' age during recruitment? (for example, >18 years of age)

If yes, then it needs to be mentioned in the methods section. And, if not, then based on what did you categorize them as young age?

We would like to thank Reviewer #1 for this thoughtful comment. We included participants between 18-35 years of age, which we included in the Methods section. (See Page 5 Line 101)

3- I would recommend adding a flow chart highlighting the recruitment process and the number of individuals included/excluded with reasons.

We fully agree with Reviewer #1 on the need of transparency in terms of the recruitment process. We now included Fig. 1. highlighting this. (See Page 6 Line 122)

4- Please replace reference 29 with a more updated/recent one.

We now included a more recent Reference to Lunn WR et al.

5- Please improve the quality of the figures provided.

The figures are now saved in full resolution. 

6- In the introduction section: "The acute changes induced..............blood supply." Please add references that support each of the claims provided.

We thank Reviewer #1 for this important comment and now included references to these sentences. 

7- In the conclusion section, both in the main text and in the abstract, please restrict your conclusions to the young age population.

We revised the conclusion of the abstract and the manuscript text to highlight this. (See Page 3 Line 51 and Page 22 Line 441)

8- Please remove the company name of the SD-OCT device from the abstract section.

We removed the company name of the OCT device from the abstract.

9- The English structure of the manuscript needs revision and editing by a native English speaker.

We had our manuscript revised by a native English speaker to improve its language structure. 

Again, we would like to express our gratitude for the important points raised by Reviewer #1 and for his efforts to improve our work.

Reviewer #2: The authors describe a prospective study of the effect of exercise on the thickness of retino-choroidal layers in twenty-one eyes (four excluded due to poor data quality), in a population of young adults.

Retina-choroidal thickness was recorded in the 9 ETDRS regions, and was measured using OCT. Image analysis and segmentation was performed using custom software. Correlation was analysed between retina-choroidal thickness pre and post exercise. Other biometric parameters (height, weight, body mass index, blood pressure, heart rate, and refraction) were considered. The authors concluded that there was retinal thinning followed by thickening post exercise, no significant correlation between the retinal and choroidal changes were detected. Two mechanisms are discussed: circulatory auto-regulatory mechanisms and a mechanical effect of IOP on retinal thickness.

Comments:

The study was generally well designed, the statistical analysis was sound and the literature adequately reviewed. The omission of IOP measurement limits the interpretability of the findings particularly when the authors conclude that this may be one of the mechanisms that could explain the study findings and this factor is easily measured.

The article is technically sound, The statistical tests have been applied rigorously, the authors have made the data available for analysis and the manuscript is written in intelligible standard.

We thank Reviewer #2 for his positive comments on our work. Indeed, the omission of IOP measurements is a limiting factor; however, we are dealing this in the most transparent fashion, encouraging the potential Readership of our paper to include this in future measurements. At the time of our study we had no technical possibility (i.e. an iCare device) that would have allowed the IOP measurements acutely after physical exercise, as our focus lied on the OCT assessments. We emphasized this point in the Discussion section of our manuscript. (See Page 22 Line 424: “however, we could not measure IOP after (and during) the physical load/strain as it would have been technically very demanding parallel with OCT imaging, the latter being of primary importance to us in this study.”) 

Minor recommendations:

Although the authors have used the standard conventions for abbreviations, the number of the abbreviations limits the readability, perhaps if a list of abbreviations could be provided in the end of the article.

We agree with Reviewer #2; however, the format of PlosOne does not allow for the inclusion of an abbreviation list.

We would like to thank Reviewer #2 for his time and efforts to improve our manuscript.

---

## [Decision Letter · Decision Letter 1]

7 Mar 2022

PONE-D-21-23138R1The assessment of acute chorioretinal changes due to intensive physical exercise in young adultsPLOS ONE

Dear Dr. Somfai,

Thank you for submitting your manuscript to PLOS ONE. After careful consideration, we feel that it has merit but does not fully meet PLOS ONE’s publication criteria as it currently stands. Therefore, we invite you to submit a revised version of the manuscript that addresses the points raised during the review process.Please ensure that your decision is justified on PLOS ONE’s publication criteria and not, for example, on novelty or perceived impact.

We look forward to receiving your revised manuscript.

Kind regards,

Andrzej Grzybowski

Academic Editor

PLOS ONE

Reviewers' comments:

Reviewer's Responses to Questions

**Comments to the Author**

1. If the authors have adequately addressed your comments raised in a previous round of review and you feel that this manuscript is now acceptable for publication, you may indicate that here to bypass the “Comments to the Author” section, enter your conflict of interest statement in the “Confidential to Editor” section, and submit your "Accept" recommendation.

Reviewer #2: All comments have been addressed

Reviewer #3: All comments have been addressed

Reviewer #4: All comments have been addressed

2. Is the manuscript technically sound, and do the data support the conclusions?

Reviewer #2: Yes

Reviewer #3: Partly

Reviewer #4: Yes

3. Has the statistical analysis been performed appropriately and rigorously? 

Reviewer #2: Yes

Reviewer #3: No

Reviewer #4: Yes

4. Have the authors made all data underlying the findings in their manuscript fully available?

Reviewer #2: Yes

Reviewer #3: Yes

Reviewer #4: Yes

5. Is the manuscript presented in an intelligible fashion and written in standard English?

Reviewer #2: Yes

Reviewer #3: Yes

Reviewer #4: Yes

6. Review Comments to the Author

Reviewer #2: The authors have addressed the comments. The analysis is technically sound and the conclusions are reasonable.

Reviewer #3: The idea of the study is interesting and I congratulate the authors for investigating the effect of exercise on OCT measurements

However there are some importante points that needs to be clarified in the study.

The authors indicate that 25 subjects were selected for the study. However, only 21 we qualified for the analysis and that is not clear in the abstract section.

Very importante is that the sample (21 eyes) seems very small for investigating so many variables. The authors evaluated 12 thickness parameters at 4 macular regions. Therefore a total of 28 analysis were performed. Also such analysis were performed at baseline and a t 5 different time periods. Because of the large number of measurments and the small sample (21 eyes) the use of a p value of 0.5 seems inadequate to me. Although I understand that using Bonferroni’s correction would be too restrictive, the use of a p value of 0.05 with so many parameters and analysis would certainly lead to significant findings by chance alone. The authors should address that issue and use a more restrictive p value.

In table 3 the authors investigate the correlation of several thickness parameters with parameters age, SE, height, weight, BMI, BDP (our DBP, diastolic blood pressure) and Heart rate. I believe the use of person’s correlation would require that all parameters adhered to normality. The authors did not mention it by did the parameters adhere to normality? It seems unlikely to me.

Also I believe evaluating the correlation between retina and choroid parameters with Age, SE, height, weight, BMI, DBP and HR is beyond the scope of the study and would require a larger number of subjects for obtaining representative results.

Data in Table 4 is also difficult to understand. Correlation between performance (PWR) and layer thickness changes ?? Did such parameters adhered to normality? What is the point of assessing such correlation? The large number of correlations assessed would easly explain obtaining some significant results (using a p level of 0.05). Since this is an exploratory investigation it would suffice to evaluate the measurements at diferente periods of time after exercise. I believe it is difficult to understand what is the meaning of the evaluation and its results.

The authors compared the results of professionals and amateurs but the number of eyes in each group seems too small to make an adequate comparison.

Reviewer #4: I want to congratulate the authors. It is a well written atricle with an interesting topic. I have some issues

1-It is known that choroidal thickness and blood flow regulation may be altered in acute and chronic smokers, so it is better to indicate if the participants are acute or chronic smokers or not

2-Exclusion criteria will be defined more compherensive and detailed

3-What did the authors do to standardize the intensity of the exercise between participants

4-Were participants asked to alcohol or caffeinated drinks consumption or food or liquid ingestion before the exercise

7. PLOS authors have the option to publish the peer review history of their article (what does this mean?). If published, this will include your full peer review and any attached files.

Reviewer #2: No

Reviewer #3: No

Reviewer #4: **Yes: **Onur Polat

---

## [Author Response · Author response to Decision Letter 1]

21 Apr 2022

We would like to thank all the reviewers for their insightful and positive comments. Please find below our detailed comments for all points raised in the reviews.

Reviewer #2: 

The authors have addressed the comments. The analysis is technically sound and the conclusions are reasonable.

Authors’ response: We thank the Reviewer for his positive feedback.

Reviewer #3

However there are some importante points that needs to be clarified in the study.

The authors indicate that 25 subjects were selected for the study. However, only 21 we qualified for the analysis and that is not clear in the abstract section.

Authors’ response: We thank the Reviewer for raising this important point that can indeed be confusing. Our Figure 1. shows the participant inclusion flowchart according to the STROBE guidelines. We included indeed 25 patients for the assessment of the demographic data; however, four subjects had missing OCT data and for this reason we excluded them from the OCT data assessment. Thus, we are reporting the data of 21 subjects, accordingly. We corrected this and made this clear throughout the entire text, including the abstract and also the Methods and Results section of the manuscript, in order to avoid confusion. (See Page 2 Line 27-28, Page 3 Line 48, Page 5 Line 101,103 and Page 10 Line 214)

Very importante is that the sample (21 eyes) seems very small for investigating so many variables. The authors evaluated 12 thickness parameters at 4 macular regions. Therefore a total of 28 analysis were performed. Also such analysis were performed at baseline and a t 5 different time periods. Because of the large number of measurments and the small sample (21 eyes) the use of a p value of 0.5 seems inadequate to me. Although I understand that using Bonferroni’s correction would be too restrictive, the use of a p value of 0.05 with so many parameters and analysis would certainly lead to significant findings by chance alone. The authors should address that issue and use a more restrictive p value.

Authors’ response: We agree with Reviewer #3 that the large number of variables and the small number of subjects may skew the results which was highlighted in the weaknesses of the study in the discussion where the pilot experimental nature of our work has also been emphasized. In line with the advice of our professional statistician, our original concept was to present the data as they are due to the controversy in the international statistical community regarding the adjustments of significance levels. Nevertheless, to improve the presentation of our results we have modified the significance level to 0.001 (see Page 2 Line 35 and Page 10 Line 208) and restructured the entire text accordingly. We now present results between 0.05 and 0.001 as “missed significance” to help the Reader in the critical interpretation of our results; we decided to include these missed significant results in order to support the planning of future studies in the field regarding potential confounders. 

In table 3 the authors investigate the correlation of several thickness parameters with parameters age, SE, height, weight, BMI, BDP (our DBP, diastolic blood pressure) and Heart rate. I believe the use of person’s correlation would require that all parameters adhered to normality. The authors did not mention it by did the parameters adhere to normality? It seems unlikely to me.

Authors’ response: We completely agree with Reviewer #3 on the importance of normality testing prior to performing statistical assessments. Indeed, according to this we tested the normality of our data by Shapiro-Wilk test which can be found both on Page 10 Line 197 in the manuscript (“The Shapiro-Wilk test was used for normality testing. For normally distributed variables parametric tests were used and continuous data are reported as mean and standard deviation”). 

Also I believe evaluating the correlation between retina and choroid parameters with Age, SE, height, weight, BMI, DBP and HR is beyond the scope of the study and would require a larger number of subjects for obtaining representative results.

Authors’ response: Indeed, we conducted a relatively small study but we do believe that highlighting these correlations does make sense for the presentation of our results and to support the planning of future studies in the field. This weakness, among others, is now addressed in the Discussion section of the manuscript. (See Page 21 Line 423.) 

Data in Table 4 is also difficult to understand. Correlation between performance (PWR) and layer thickness changes ?? Did such parameters adhered to normality? What is the point of assessing such correlation? The large number of correlations assessed would easly explain obtaining some significant results (using a p level of 0.05). Since this is an exploratory investigation it would suffice to evaluate the measurements at diferente periods of time after exercise. I believe it is difficult to understand what is the meaning of the evaluation and its results.

Authors’ response: We thank the Reviewer for this valuable comment. We assessed the correlation between performance and the change of the layer thickness in order to assess whether the extent of physical activity influences the observed changes. The normal distribution of the data was checked also in this case (see our previous response). We are presenting our data in the revised manuscript based on the 21 subjects and using the modified p value of 0.001 as suggested by the Review. (See above.) (For the changes see Page 16, line 307-318)

The authors compared the results of professionals and amateurs but the number of eyes in each group seems too small to make an adequate comparison.

Authors’ response: We are grateful to the Reviewer for raising this point. Our primary goal with this pilot study was the assessment of retinal changes in young adults. As the participants had different backgrounds regarding professional sports activities, we found it appealing to possibly compare or at least describe these two subgroups. We agree with the Reviewer that such a comparison would deliver questionable results and, therefore, already included it in the original Discussion of the study weaknesses (see Page 21 Line 422). We would prefer to leave it up to the Reader’s decision how they interpret these findings. 

Reviewer #4:

I want to congratulate the authors. It is a well written atricle with an interesting topic. 

Authors’ response: We thank the Reviewer for his positive feedback.

I have some issues 1-It is known that choroidal thickness and blood flow regulation may be altered in acute and chronic smokers, so it is better to indicate if the participants are acute or chronic smokers or not 

Authors’ response: We would like to thank the Reviewer for this important comment. None of the subjects were smokers which we additionally included in the Results. (See Page 10 Line 215)

2-Exclusion criteria will be defined more compherensive and detailed

Authors’ response: We have now all the exclusion criteria listed in the text (Page 6 Line 118)

3-What did the authors do to standardize the intensity of the exercise between participants

Authors’ response: Thank you for raising this excellent point in the review. The way how intensity was standardized lies actually in the way we increased the loading itself, as originally described in the methods (Page 7 Line 131): each participant performed a stepwise incremental exercise trial until exhaustion (vita maxima) on a rowing ergometer. The intensity has been increased every 500 meters with the next load step required to be performed 10 seconds faster, until complete fatigue. The maximum power achieved in the last load step was considered in the evaluation. In order to further standardize the exercise intensity and make the values comparable among participants, for the assessments we used a calculated power-to-weight ratio (PWR, expressed in watt/kg). We believe this methodology enables a relatively unbiased comparison of the data.

4-Were participants asked to alcohol or caffeinated drinks consumption or food or liquid ingestion before the exercise

Authors’ response: We are grateful to the Reviewer for this valuable comment. None of the subjects drank alcohol or caffeine at least 24 hours before the exercise. The meals were not prescribed; however, all subjects had had their regular breakfast at least one hour before the exercise. We added this information to the Methods section of the manuscript. (See Page 7 Line 128.)

---

## [Decision Letter · Decision Letter 2]

9 May 2022

The assessment of acute chorioretinal changes due to intensive physical exercise in young adults

PONE-D-21-23138R2

Dear Dr. Somfai,

We’re pleased to inform you that your manuscript has been judged scientifically suitable for publication and will be formally accepted for publication once it meets all outstanding technical requirements.

Kind regards,

Andrzej Grzybowski

Academic Editor

PLOS ONE

Additional Editor Comments (optional):

Thank you for conducting this interesting study and submitting your paper to our journal.

Reviewers' comments:

Reviewer's Responses to Questions

**Comments to the Author**

1. If the authors have adequately addressed your comments raised in a previous round of review and you feel that this manuscript is now acceptable for publication, you may indicate that here to bypass the “Comments to the Author” section, enter your conflict of interest statement in the “Confidential to Editor” section, and submit your "Accept" recommendation.

Reviewer #2: All comments have been addressed

Reviewer #4: All comments have been addressed

2. Is the manuscript technically sound, and do the data support the conclusions?

Reviewer #2: Yes

Reviewer #4: Yes

3. Has the statistical analysis been performed appropriately and rigorously? 

Reviewer #2: Yes

Reviewer #4: Yes

4. Have the authors made all data underlying the findings in their manuscript fully available?

Reviewer #2: No

Reviewer #4: Yes

5. Is the manuscript presented in an intelligible fashion and written in standard English?

Reviewer #2: Yes

Reviewer #4: Yes

6. Review Comments to the Author

Reviewer #2: The authors have performed the necessary alterations in the manuscript. The analysis is technically sound and the authors have correctly outlined the limitations of their study.

Reviewer #4: (No Response)

7. PLOS authors have the option to publish the peer review history of their article (what does this mean?). If published, this will include your full peer review and any attached files.

Reviewer #2: No

Reviewer #4: **Yes: **Onur Polat

---

## [Editor Report · Acceptance letter]

16 May 2022

PONE-D-21-23138R2 

The assessment of acute chorioretinal changes due to intensive physical exercise in young adults 

Dear Dr. Somfai:

I'm pleased to inform you that your manuscript has been deemed suitable for publication in PLOS ONE. Congratulations! Your manuscript is now with our production department. 

Kind regards, 

on behalf of

Dr. Andrzej Grzybowski 

Academic Editor

PLOS ONE